# The Future of Essentially Derived Variety (EDV) Status: Predominantly More Explanations or Essential Change

**John Stephen C. Smith**

Department of Agronomy, Iowa State University, 2104 Agronomy Hall, 716 Farm House Lane, Ames, IA 50011-1051, USA; jscsmith@iastate.edu

**Abstract:** This review examines the categorization of Essentially Derived Varieties (EDV) introduced in the 1991 revision of the Convention of the *Union internationale pour la protection des obtentions végétales* (UPOV). Other non-UPOV member countries (India, Malaysia, and Thailand) have also introduced the concept of essential derivation. China, a UPOV member operating under the 1978 Convention, is introducing EDVs via seed laws. Challenges in the implementation of the concept and progress made to provide greater clarity and more efficient implementation are reviewed, including in Australia and India. The current approach to EDV remains valid provided (i) clarity on thresholds can be achieved including through resource intensive research on an individual crop species basis and (ii) that threshold clarity does not lead to perverse incentives to avoid detection of essential derivation. However, technological advances that facilitate the simultaneous introduction or change in expression of more than "a few" genes may well fundamentally challenge the concept of essential derivation and require a revision of the Convention. Revision could include deletion of the concept of essential derivation coupled with changes to the breeder exception on a crop-by-crop basis. Stakeholders might also benefit from greater flexibility within a revised Convention. Consideration should be given to allowing members to choose if and when to introduce changes according to a revised Convention on a crop specific basis.

**Keywords:** intellectual property; intellectual property protection; plant variety protection; plant breeders' rights; essentially derived variety; utility patent; plant breeding; biotechnology

## 1. Introduction

The ability to obtain protection of newly developed plant varieties as intellectual property (IP) can encourage investments into plant breeding [1]. The World Trade Organization's (WTO) Agreement on Trade-Related Aspects of Intellectual Property (TRIP) requires a patent or "effective sui generis" plant variety system (Article 27.3(b)). The most globally used form of IP, and one that is deemed acceptable by the WTO, is the sui generis or specially developed approach to Plant Variety Protection (PVP) or Plant Breeders' Rights (PBR) that was introduced under the auspices of the *Union internationale pour la protection des obtentions végétales* (UPOV), or International Union for the Protection of New Varieties of Plants, and adopted in 1961 [2]. The mission of UPOV is: "To provide and promote an effective system of plant variety protection, with the aim of encouraging the development of new varieties of plants, for the benefit of society" [3]. Other sui generis forms of PVP have been adopted by countries that are not members of UPOV. These include Thailand, Malaysia, and the Protection of Plant Variety and Farmers Right Act (PPVFRA) in India.

The fields of plant breeding and associated biotechnologies have witnessed major changes in technological capabilities since the initial UPOV Convention in 1961, although their adoption varies across crops and regions. Such changes inevitably subject a sui generis IP system to pressure for adaptive change in order to maintain encouragements to undertake research and product development, their very raison-d'être [4]. An example of such change was the introduction of an Essentially Derived Variety (EDV) in the 1991

Convention [5]. Provision for EDVs has also been included in PVP legislation by non-UPOV members including India, Malaysia, and Thailand. Whilst China is an UPOV member operating under the 1978 Act, provisions of EDV according to UPOV 1991 are being adopted in their revision of seed laws. The PPVRA of India is unique among global PVP laws allowing applicants to apply for protection as an EDV per se [6].

Determination of essential derivation is highly dependent upon data that can inform a decision on the degree to which, (i) one variety is inherited or derived from another, and (ii) apart from changes introduced during further development, the extent to which essential characteristics of the initial variety have been retained in the derivative(s). A sound basis of technical information can both contribute to and help inform a decision on essential derivation. However, increased technological capabilities to transfer or change the expression of several genes simultaneously may further question the degree of genetic change and/or breeding effort required to warrant independent commercial status. Consequently, it is appropriate to question whether the introduction of essential derivation in 1991 remains a practical solution to support a balance of IP rights between initial and subsequent developers of new varieties. A review is timely because UPOV is in the process of hearing presentations from stakeholders during the preparation of a third set of explanatory notes on essential derivation [7–9]. This review examines the effectiveness of EDV to date, considers lessons learned and alternate proposals that have been made. The review concludes with identified conditions under which further revision of the sui generis approach would be warranted and includes major elements of such revision.

## 2. The UPOV Approach to PVP

Additional information on the evolution and implementation of the UPOV system is available at https://www.upov.int/portal/index.html.en (accessed on 20 June 2021) and [10–13]. The UPOV system provides for PVP or PBR. These rights prevent unauthorized copying or repeated use of protected varieties during a time-limited period of protection. However, further breeding using the protected variety and subsequent commercialization by non-title holders is allowable. PVP effectively provides a comprehensive open-source system for further breeding of self-pollinated crops and hybrids. However, the public availability of parental lines of hybrids following expiration of their protection is not a UPOV requirement but is provided for by the USDA after expiration of IP in the USA.

As of 22 February 2021, UPOV has 77 members [14]. The basic technical requirements to obtain a PVP have remained unchanged since the origin of UPOV in 1961 to today:

1.  Distinctness (distinct from all other publicly known varieties),
2.  Uniformity (with respect to and in accordance with biology), and
3.  Stability (maintain its distinct characteristics during reproduction from generation to generation); collectively known as the DUS criteria.

UPOV 1991 introduced two major changes to the scope of protection previously afforded: (i) scope extending to harvested produce and (ii) scope of rights to commercialize without consent by the owner of an initial variety from which a variety deemed to have been essentially derived was developed. The ramifications of the former change have been discussed [13]. The focus of this review is upon the latter, the concept of essential derivation. Following a review of the rationale that led to its introduction, the discussion moves to the challenges of implementation, consideration of other options, and a recommendation for future change. Readers are directed to Helfer [15] for a detailed comparison of UPOV 1978 and UPOV 1991.

## 3. The Concept of Essential Derivation

*3.1. The Need and Rationale for the Introduction of Essential Derivation and the Category of an Essentially Derived Variety (EDV) in the UPOV 1991 Convention*

The field of biotechnology began from basic research during the 1830s [16], which continued during the first decade of the 20th century, but did not begin in earnest until the 1960s [17]. The first transgenic varieties were introduced into cultivation during the

1990s and included tomatoes with delayed ripening and field crop varieties endowed with insect resistance and herbicide tolerances [16–18]. Prior to the release of transgenic varieties there were relatively few examples of single genes of positive economic effect that were not available on equal terms, including via the public domain to most, if not all breeders, e.g., native disease resistance genes. Consequently, the introduction of publicly available genes through crossing of the trait donor genotype and recipient variety with selection for the desirable trait did not disrupt or bias the level of IP among breeders afforded under UPOV 1978, regardless of their respective technological or economic capacities.

Subsequent advances in the field of biotechnology enabled a new category of simply inherited traits sourced from other phyla that provided resistances to insects and to herbicides, following their transgenic insertion into cultivated plants. These traits are of great economic importance, strongly protected by utility patents, and available under license from a small cadre of well-resourced and technologically enabled organizations. The introduction of these traits via transformation into varieties protected under UPOV 1978 would have provided an "enabling advantage" to developers using molecular tools compared to others who solely used "essentially biological," i.e., crossing and selection methodologies [12,19–23]. Consequently, remedial IP treatment was required to create a more equitable sui generis system, one that provided encouragement to undertake both crossing and selection and the development and integration of new "biotechnology" traits. The solution adopted by UPOV 1991 exemplified a balance-based approach. This balance was achieved by the extension to developers of an initial variety (iv), the rights to control commercialization of a progeny variety that was deemed to be essentially derived, i.e., an Essentially Derived Variety (EDV). Further details on the EDV concept have been provided, including [1,12,23–32]. The inclusion of essential derivation in the PPVFRA of India is one component of a more comprehensive approach to the provision of IP for plant varieties through an ability to also protect "extant" varieties [33]. Consequently, the goal of including essential derivation in the PPVFRA is to enable balanced protection among all developers of improved crop varieties, including farmer-breeders due to the large number of such varieties in cultivation in that country.

*3.2. Challenges That Have Arisen in the Determination of a Variety Categorized as Being Essentially Derived*

The most challenging aspects facing practical implementation are contained within Article 14(5)(b) of the 1991 Act of UPOV, with emphasis placed by this author on specific wording using bolded text:

"a variety shall be deemed to be essentially derived from another variety (the initial variety) when:

1.  it is **predominantly derived** from the initial variety, or from a variety that is itself predominantly derived from the initial variety, while retaining the expression of the **essential characteristics** that result from the genotype or combination of genotypes of the initial variety,
2.  it is **clearly distinguishable** from the initial variety, and
3.  except for the differences which result from the act of derivation, it conforms to the initial variety in the expression of the essential characteristics that result from the genotype or combination of genotypes of the initial variety."

UPOV has sought to provide clarification, including for three critical issues [34,35], with emphasis placed by the author of this review using bolded text:

1.  Predominantly derived: "a variety can only be essentially derived from **one initial variety**", i.e., a direct parent–progeny relationship by pedigree and "a variety should only be essentially derived from another variety when it retains **virtually the whole genotype** of the other variety."
2.  Essential characteristics: "includes **all heritable traits** that contribute to the principal features, performance or value of the variety; from the perspective of the producer,

seller, supplier, buyer, recipient, or user; and essential characteristics may be different in different crops/species."

3. Degree of difference to be within the EDV boundary of an initial variety: "be different from that variety by a very limited number of characteristics."

The degree of difference requirement per se places no upper threshold on the number of differences that result from the act of derivation other than in respect to:

1. a limitation imposed by retaining virtually the whole genotype of the other variety and so reinforcing the enabling advantage to those using molecular tools compared to those who solely used essentially biological, i.e., crossing and selection.

2. the examples given in Article 14(5)(c), which comprise a non-exhaustive list: "the selection of a natural or induced mutant, or of a somaclonal variant, the selection of a variant individual from plants of the initial variety, backcrossing, or transformation by genetic engineering" ... which "make clear that the differences which result from the act of derivation should be one or very few."

To summarize, designation of a variety as an EDV requires it to be distinct, a progeny in a pedigreed relationship to the iv, to retain virtually the whole genotype of the iv, while differing from the iv by no more than a very limited number of characteristics. Those characteristics include all those that are heritable and/or of economic value to stakeholders, from producers to users.

### 3.3. Experiences during the Implementation of UPOV 1991 to Date and Lessons Accrued

I am indebted to an anonymous reviewer of an early draft of this manuscript for their observation that for most court cases [36–40], judicial decisions have gone against parties who bear the burden of proof. Such an imbalance in outcomes might indicate problems in evidential testimony or the requirement of an overly high burden of proof. A review of the use of molecular marker data in several of these cases indicates that a major problem lay in the generation and presentation of those data to the courts. In [38], which involved varieties of the genus *Gypsophila*, the Court of Appeal was "open to objections" regarding the use of the specific marker technology of amplified fragment length polymorphisms (AFLPs) [41]. Objections were that: (i) AFLP technology generates a dominant or mono-allelic markers system, (ii) markers were not evenly present through the genome of the crop species, and (iii) there was no presentation of measurement error. Consequently, the Court was concerned that: (i) some genetic diversity might not be represented, (ii) some chromosome regions were not sampled, and (iii) there was no statistical basis to determine the degree of significance in the measurement of genetic similarity. The specific issues relating to AFLP technology were well-known at the time, yet had not detracted from the routine use of that technology in applied plant breeding programs.

In contrast, AFLP data were presented to and accepted as credible evidence by the Israeli court [39]. However, presentation of AFLP data by the plaintiffs to the Israeli court included analyses demonstrating the capability of these data to show that the initial variety and putative EDV shared high genetic similarity within the context of their association within a larger set of *Gypsophila* varieties. In other words, these data demonstrated to the court the discriminative capability of AFLP data among varieties of *Gypsophila*. However, AFLP data presented by the defendants contradicted those presented by the plaintiffs. Nonetheless, the court was not distracted by this potential confusion following testimony by the defendant that the initial pedigree provided by them was in error [42]. Issues relating to the use of AFLP technology have since become largely moot due to the rapid evolution of marker technologies. Today, other marker technologies can allow more than one million single nucleotides to be assayed for individual crop species. Consequently, unless cultivars of a crop species have been developed from an exceedingly narrow genetic base, then currently developed marker systems are available to be used to measure genetic similarities for the purposes of helping to determine essential derivation. There is a huge body of literature reporting usage of molecular marker data in respect to EDV. The citations [43–50] were selected to provide specific focus with regard to the use of marker data as evidential

material to help in determining EDV status. They also provide an entry point to the broader literature reporting research on this subject.

Examination of litigation [36–40] has also shown the importance of pedigree data in helping to resolve essential derivation. The lack of pedigree data is, at the very least, highly problematic for the developer of an alleged EDV. For example, information on: "differences which result from the derivation is possible only through facts available exclusively to the person claiming to be the breeder of (a second) variety. Only he knows how the new variety was achieved" [51]. Consequently, "In the pertinent submission it will be necessary to demonstrate in detail which breeding program was used and how the process was applied. The frequent assertion by infringers, in particular in cases of vegetatively propagated plants, that the new variety resulted from seedlings of their own plant material, would not suffice" [51].

The Community Plant Variety Office (CPVO) of the European Union noted that Courts have generally seemed to accept that a showing of high genetic conformity should reverse the burden of proof [52]. This rationale is sensible because the developer of the putative EDV has access to most of the relevant information required for the determination of EDV status [52,53]. The chances of developing an EDV have either been avoided or at least reduced by choice of initial germplasm, or by selection of progeny that might not reasonably be considered predominantly derived. The upfront negotiation of licenses avoids risks, uncertainties, potential costs from lawsuits, and delays in commercialization. Most EU holders of PVP ivs and PVP EDVs have agreed business-based solutions [52]. This author does not accept the argument that having to seek agreements up front places undue burdens on the second developer [54]. For if a particular variety or varieties represent the optimum technical and/or commercial base–germplasm choice(s) for a later developer to access, then the worth of that specific genetic base deserves respect of its embodiment as IP. In contrast, if publicly available germplasm is equally or more desirable as a source material for a later developer, then the need for an up-front agreement is moot and there is no prospect of developing an EDV.

*3.4. Technical Practices Involved in the Implementation of Essential Derivation According to the PPVFRA of India*

Experts in India face the same challenges in determining EDV status as do breeders in UPOV member countries. For example, the ubiquitous question of "how far an EDV and the original variety have to resemble each other phenotypically is a difficult one to answer, since the definition offers scope for various interpretations" [33]. Breeding records and pedigree information are prerequisites to resolve disputes. "If a defendant is unable to provide detailed and accurate pedigree records that are subsequently verified, then this omission could disqualify claim to a variety" [33]. Determination of essential derivation is optimally based upon genotypic data rather than solely on phenotypic comparisons. However, the availability of suitable molecular marker data only extends to a few crops. As of 2009 [33], procedures and protocols for the determination of essential derivation in rice, wheat, maize, and pearl millet were being developed by the Division of Genetics within the Indian Agricultural Research Institute (IARI). Current applications for protection as an EDV rely on the compilation of as much useful data as possible including breeding records, comparisons of morphological data also used in DUS between the initial variety, the putative EDV, and reference varieties. Expectations are that a derived variety that differs by at least one DUS characteristic, which otherwise retains high conformity to the iv, and expresses most of the essential characteristics of the iv, will be categorized as essentially derived from that iv. In the case of dispute, the burden of proof is upon the holder of protection of the iv. If the holder of IP for the derived variety disputes its categorization as essentially derived, the holder of IP of the iv may request intervention by the authority or a methodology and national court. All forms and a technical questionnaire that are required to be completed with an application of protection as an EDV are published [6].

**4. Recent Questions and Concerns That Have Arisen Leading toward the Development of a Third Set of UPOV Explanatory Notes on EDVs**

*4.1. Who Decides?*

It was never envisioned that determination of varietal status as an EDV would be made by examining offices but rather by plant breeders through mutual agreement, or failing that, as a result of litigation [55].

*4.2. Use of Partial UPOV Text Leading to Determinations Inconsistent with the Language and Intent of UPOV 1991*

Interpretations that an EDV must include **all** the essential characteristics of an iv (emphasis by author) are not compliant with UPOV 1991. For Article 14(5)(b) (iii) of the 1991 Act reads (bolded text emphasizes by author):

(iii) "except for the differences which result from the act of derivation, it conforms to the initial variety in the expression of the essential characteristics that result from the genotype or combination of genotypes of the initial variety."

The exclusion of the part of Article 14(5)(b) (iii) that is bolded above when written into national PVP laws, either reads or is interpreted as an EDV must "conform to the initial variety in the expression of all the essential characteristics". However, using the broad definition of "essential" in this context, as interpreted by UPOV [35], then retention of all characteristics is a biological impossibility. For when a distinct new variety is developed by either the addition of a new characteristic or through a change in expression of an existing characteristic, then it cannot also retain all the essential characteristics of the iv. For example, to add disease resistance to an initial variety means that the derived variety no longer retains the essential characteristic exhibited by the iv of susceptibility to that disease. Hence, use of the language in UPOV 1991: "except for the differences which result from the act of derivation".

Additional questions arise when characteristics are categorized as either essential or non-essential according to a qualitative assessment of their agronomic or economic importance. If a qualitative distinction is made, then the following questions and added uncertainties arise, including: Who makes the determination, and how is the determination made?, what, if any, degree of difference in expression is sufficient to designate a previously non-essential characteristic as now being essential?, what, if any, degree of difference in expression is sufficient to make a trait even more essential than it previously was? This reviewer understands that such distinctions can contribute improvements to an IP system. However, such changes may also require a more fundamental overhauling of the IP system itself.

Determination of EDV status by the Australian PVP office is made according to such a qualitative interpretation of their essentiality. If a later developed variety only differs from an iv by non-essential characteristics, then it is deemed to be an EDV, and its commercialization is controlled by the developer of the iv. This approach is positive in that it effectively removes varieties from commercialization that are plagiaristic and/or the result of "cosmetic breeding". It can also have a positive outcome in ensuring the commercial availability of an improved variety. However, a significant problem also arises from this narrow interpretation of essential derivation. For when a derived variety differs from the iv for an "essential" characteristic, then it is deemed by the Australian PVP office not to be an EDV. The derived variety, which might have been determined to be essentially derived under other circumstances, can be freely commercialized by the later developer. This approach contradicts the very raison-d'être of the balanced approach of providing IP rights to the initial breeder and subsequent developer introduced by the UPOV 1991 Convention. This rationale of determining essential derivation thereby runs the risk of undermining incentives to undertake crossing and selection to improve a more comprehensive array of quantitatively and qualitatively inherited traits in favor of making small genetic changes to existing varieties.

### 4.3. Persistent Questions and Additional Recent Concerns Regarding Determination of Essential Derivation

Persistent challenges noted above together with concerns related to the rapid acceleration of technologies have led to the current preparation by UPOV of a third set of explanatory notes on EDVs. Concerns come from several sources. First, developers who make significant use of crossing and selection have expressed concerns that advances in capabilities to rapidly modify the phenotypic expression of several genes may undo the current balance of providing IP to initial and subsequent developers of improved varieties. Second, there are concerns by those who are increasingly technically enabled to modify phenotypic expression of several genes rapidly and, in parallel, that guidelines for essential derivation may undermine their future willingness to undertake research and product development. Among the former, the greatest concerns are expressed by developers of asexually propagated varieties. Single gene mutants provide the source for many new varieties of asexually, or vegetatively reproduced crops. New varieties of these crop species can also be developed through the sexual cycle, which creates new diversity yet then requires long-term selection to identify improved segregants and recombinants. A concern being that improved varietal selections then have the potential to be "captured" by the making of one or a very few additional mutations. Consequently, there are concerns that there may be an escalation of EDV-related disputes in the near future.

Consequently, there is an immediate focus on developing greater clarity through a further revision of EDV explanatory notes. Alternate approaches have also been proposed. These are presented and commented upon below in their order of increasing departure from the current state.

## 5. Major Proposals That Have Been Made for the Revision or Elimination of the Concept of Essential Derivation

### 5.1. White Paper on "Essentially Derived Varieties"

Clearly, decision makers, including of course the judiciary, will be those who make a determination of essential derivation. A comprehensive set of technical information optimally provides evidential material that can be drawn upon to help in the determination of EDV status. However, proposals that rely primarily on phenotype with use of a compulsory license [56] are overly limited. This approach is flawed by (i) requiring all essential characteristics to be retained, (ii) removing the initial prerequisite of determining predominant derivation, and (iii) being imbalanced by placing an over-reliance upon phenotypic compared to genotypic data.

It is technically incorrect to assert that "Absolute measures of genetic similarities are not scientifically feasible" [56]. However, the assertion that "Quantitative thresholds have to be constantly monitored in order to comply with the innovations concerning new breeding technologies" [56] is certainly true and underlines the immensity of technical effort required to establish sound technical guidelines. This reviewer concurs that: "A juridical approach is dynamic, as it can adapt to the evolution of plant breeding practices and variety production" [56]. However, the quality of any resulting decision, whether it be made through mutual agreement or in the courts, is surely highly dependent upon a sound evidential foundation. The use of compulsory licenses is not supported by the global seed industry [53].

### 5.2. A 4 Pillar Approach

This approach [54] is based upon the following premises:

1.   Characteristics can be categorized according to whether they exhibit essential characteristics or not;
2.   The addition or change of a characteristic that represents added value automatically results in the derived variety being outside the realm of an EDV;
3.   Change that is non-essential or insubstantial is plagiaristic and therefore an EDV.

Each premise is problematic. With regard to:

1. UPOV does not differentiate among characteristics according to a qualitative interpretation of "essential" or "value-added", and one can foresee prospects of endless arguments with poor to no legal precedent being set on the categorization of traits and their relative expression levels as being essential or adding value.

2. According to this approach, being value-added results in being outside the scope of EDV, thereby reverting to the UPOV 1978 Convention, which promotes free riding by the second developer.

3. While plagiarism can be further enforced against, it was never the motivating force that led to the introduction of essential derivation. Indeed, plagiarism or cosmetic breeding can be dealt with during the determination of distinctness, e.g., as practiced by the use of GAIA, an approach that differentially weights the relative importance of characteristics in their contribution to the determination of distinctness [57]. Equally problematic is the proposed basis for determining EDV status for varieties that are said to result from incremental breeding steps and thus according to [54] do not express essential characteristics. With regard to these crops, and according to the premises previously stated, the addition of a value-added characteristic would allow the developer of the derived variety to free ride as per the UPOV 1978 Convention.

## 5.3. Utilize the Doctrine of Equivalence

This approach is based upon essential derivation being treated as similar to the doctrine of equivalence in patent law [23]. A three-step sequential procedure is envisioned to be utilized to determine EDV status: (1) Does the derived variety achieve substantially the same result in substantially the same way as the initial variety, i.e., does it retain the essential characteristics of the initial variety? If so, then (2) would such a determination be obvious to a person skilled in the art? If so, (3) did the PVP holder of the iv intend the relevant characteristics to be an essential requirement of the application? This approach was ultimately rejected for not being practically feasible, nor relevant to the PVP system [23].

## 5.4. Free Access but Obligation to Pay Compensation for Use

This approach [23] places responsibility upon the initial developer to provide evidence that their protected variety had been used to develop another variety. Use by a third party would then trigger a "use payment" which could only be voided by the third party providing proof that use of the protected variety was not an essential component of its development, a reversal of burden of proof. However, this approach is problematic because it remains subject to all the outstanding and inherent questions and challenges underlying a determination of essential derivation status.

## 5.5. Categorization of Characteristics in Relation to Varietal Performance

This approach [58] proposed that characteristics could be categorized according to their contribution to varietal performance. For example, initial variety status would not be achieved until the contribution of one or more expressed traits provided added value. The proponents [58] considered that such an approach would be especially applicable to complex traits, such as those developed through individually measurable advances. However, the level of precision required to measure iterative progress in quantitative traits such as drought resistance or yield, would render such an approach impractical with regard to field crops. Furthermore, addition of a value-added trait allows the second developer to free ride capturing the germplasm developed by the initial breeder, thereby reverting to UPOV 1978.

## 5.6. A Special Research Exemption

This proposal [58] envisages that IP for varieties that exhibit incremental improvements would continue to reside under UPOV with no EDV regime. In contrast, varieties exhibiting "improvements of greater significance" [58] could be eligible for patent protection providing the patent regime included a statutory research exemption. A critical

problem with this approach is that most genetic gain for field crops has been reliant on, and will likely continue to depend upon, incremental advances for traits such as yield and drought resistance that are under complex genetic control. Implementation of this approach would reinstate the imbalance that led to UPOV 1991 and simply revert to UPOV 1978, thereby making developers of initial varieties vulnerable to free riding.

Others [54,59,60], have also proposed that characteristics be defined in respect of being essential or not. The essential problem remains: Who makes the definition and upon what basis? All the subsequent challenges noted above then still remain, how to define "predominant" and how to define "a few" on a species-specific basis. Such proposals cannot provide further clarity to the determination of essential derivation.

### 5.7. Revision of the Breeder Exception

This approach [61–64] proposed a revision of UPOV whereby open-source use for further breeding would not be available immediately upon commercialization but instead would be delayed by a number of years, e.g., between 3–10 years dependent upon crop species. All varieties and parental inbred lines would then be available in the public domain once their period of protection had expired; a guarantee of access that is not currently provided by the UPOV.

### 5.8. Compensation Liability Regime

This approach [65] involves radical change through the introduction of a "compensatory liability regime" (CRL) that would avoid concerns of increased speed to market by second developers at the potential detriment to the interests of and further incentives to invest by the initial developer. Under such a scheme, a second developer "would obtain a license to compete by paying to the originator a prescribed multiple of the measured investment which the original breeder had made under uncertainty and high risk. The follower would in fact be sharing in that investment and its risk retrospectively" [65]. Under such an approach it is anticipated that benefits would accrue to multiple stakeholders: (i) varieties endowed with higher economic value will attract more licensees, thereby encouraging further investments including in relatively high-risk research; (ii) all new germplasm would be accessible for further development, thereby increasing follow-on investments; (iii) when germplasm is used under the terms of the International Treaty on Plant Genetic Resources for Food and Agriculture (ITPGRFA) [66] or the Convention on Biological Diversity (CBD) [67], then a portion of license fees could accrue back to support the goals of the Treaty via the Global Crop Diversity Trust [68] and the benefit sharing fund of the Treaty or to other providers, if accessed under the CBD.

## 6. Concluding Comments and Proposed Path Forward

### 6.1. The Current and Evolving Technology Environment That Informs the Conduct and Effectiveness of Plant Breeding

The UPOV 1991 Convention was introduced to provide balanced encouragement via IP between approaches to variety improvement by (i) crossing and selection of anonymous genes and (ii) the insertion or changes of specific genes and associated regulatory sequences, including those obtained from other species. The technological environment within which UPOV 1991 was developed and introduced has continued to undergo rapid change providing additional technical capabilities and knowledge. As previously noted, several approaches have been proposed to implement or to revise essential derivation. However, most proposals either fail to resolve outstanding questions on thresholds or are otherwise equally or more problematic.

Selection upon a broad array of genomic diversity, anonymous with respect to specific genes and functionality, has contributed the vast preponderance of genetic gain for quantitative traits. Phenotypes that result from interactions among genes and expression sequences of individually small effect with environmental factors makes individual gene identification and measurement of their effects very challenging. However, such identifi-

cation is a prerequisite to allow change in the phenotypic expression of specific targeted genes. A complementary approach to increased breeding efficiency has been effected through the targeted selection of anonymous genes using marker-assisted selection (MAS) and genome-wide selection (GWS).

Genes involved in the expression of heat and cold shock proteins can modify the expression of quantitative traits. However, the progression of single-gene mutants into commercial varieties lags several decades behind simply inherited traits that provide insect resistance and herbicide tolerance. Reports citing the improved expression of drought tolerance by the modification or addition of single genes have been criticized due to a lack of rigor in sufficiently demonstrating underlying physiological and genetic mechanisms [69]. However, to date there has been a relative lack of progress in developing drought resistance through single gene modifications due in part to insufficient testing using physiological and genetic models that are biologically relevant [70]. Numerous candidate genes associated with many diverse QTL have been listed, however, most citations were at an elementary stage with a focus on yield drag [71]. Recent reviews [72,73] indicate that alterations in the expression of specific genes may contribute to the agronomic improvement of quantitative traits. Additional capabilities to simultaneously insert 10 or more additional and/or edited genes have been developed [74,75] and open-source methodologies to insert genes are available [76]. It can be anticipated that varietal improvement through the introduction of single gene changes that result in economically important phenotypes will continue [77,78]. The UPOV approach has been criticized for failing to solve a basic contradiction, that being to "provide protection for two very different forms of genetic enhancement, discrete and complex, within a single system" [58]. Further progress in capabilities to improve varieties through the addition or change to an increasing number of specific genes, including by multi-gene cassettes, will further exasperate categorizations of characteristics or traits as either "discrete" or "complex" and so render those definitions increasingly meaningless in the context of essential derivation.

An additional concern regarding implementation of EDV status stems from capabilities to pursue perverse incentives to undertake misappropriation of IP by cosmetic breeding and/or the development of EDVs while avoiding their detection. For example, increased metrical clarity on a crop specific basis for thresholds on predominant derivation and/or retainment of essential characteristics triggering potential EDV status enables more effective implementation of the concept. However, technologies could also be used to develop derivatives selected to have all the desirable performance attributes of the iv with genetic and phenotypic distances that exceed potential EDV triggering thresholds. These technologies include: (i) high through-put molecular marker laboratories, (ii) genotyping parental lines of hybrids using maternally inherited tissue [63], (iii) isolation of the female parent of a hybrid by a strategy known as "chasing selfs" [79,80], and (iv) use "reverse-breeding" [81,82] that allows reconstruction of hybrids through a "shuffling" of the initial parental genotypes.

*6.2. A Proposal*

Plant breeders must determine their relative expenditure of resources among (i) identifying and manipulating specific known genes compared to crossing and selection upon anonymous genes, and (ii) working with known well-adapted and already widely-used germplasm compared to less immediately well-adapted, including exotic germplasm. IP policies establish important parameters that inform investment decisions and influence breeding strategies. The making of germplasm access and use agreements upfront makes sound business and technical sense in circumstances when the initial variety has characteristics that are also preferentially desired by the second developer. Otherwise, subsequent developers have access to publicly available germplasm, thereby making moot the occurrence of essential derivation.

If the preceding discussion on which categories of genes might be the most precise or optimal to use in the development of improved varieties appear nonsensical, or may

be increasingly difficult to demarcate, then such conclusions support an approach to IP protection that is independent of such categorization. It is important to provide an IP regime whereby all breeding approaches can coexist in a balanced symbiotic relationship to provide optimal outcomes in terms of improved products for the benefits of all stakeholders.

This author anticipates that the concept of essential derivation will remain a valid and increasingly useful approach provided:

1. Breeding methods modify or change "a few" genes that result in differences that are readily observable and/or measurable;
2. Metric thresholds have a high degree of consensus on a crop specific basis;
3. The definition of "essential" is at least agreed on a crop specific basis;
4. Accurate and sufficiently detailed pedigree data are a prerequisite and the absence of which demonstrates culpability;
5. That attempts are not made to evade essential derivation through use of "reverse breeding" or other technologies with equivalent outcomes;
6. That molecular marker comparisons help determine predominant derivation provided that the technical basis of usage has been established on a crop specific basis;
7. Demonstration of predominant derivation causes the reversal of burden of proof to be upon the developer of the putative EDV who is best placed to provide pertinent evidential responses.

However, as technological progress continues, variety development through the simultaneous addition and/or changes in expression of more than "a few" genes will occur [73–78]. Additionally, it may be that judicial precedent on the definitions of: "a few", "predominant", and "essential" becomes problematic or contradictory in terms of supporting the advancement of genetic gain through plant breeding. Furthermore, there are significant technical challenges and resources required to establish the technical foundations needed for the degree of clarity necessary to facilitate up front agreements and/or the smooth resolution of disputes. Consequently, it may well be impossible to develop such a technical foundation for more than a few crop species.

Plant breeders and farmers thoroughly understand the enduring need for varieties that are increasingly better adapted to an environment comprised of changing and unpredictable biotic and abiotic factors. IP systems also adapt and evolve as a result of interactions with accumulated knowledge and technological capabilities. Consequently, it would be wise to consider potential elements of further evolution and adaptation of the sui generis system beyond UPOV 1991. Elements that members might consider include:

1. Deletion of the concept of essential derivation;
2. A revision of the breeder exception on a crop specific basis so that open access for further breeding is delayed on a crop specific basis;
3. Allow breeders to provide access under mutually agreed terms before open access is provided, and
4. Ensuring access to varieties and parental inbred lines once their period of protection has expired.

Stakeholders might also benefit from greater flexibility within a revised UPOV Convention. Consideration should be given to allowing members to choose if and when to introduce changes according to a revised UPOV Convention on a crop specific basis.

**Funding:** This research received no external funding.

**Conflicts of Interest:** The author declares no conflict of interest.

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
