# Peer review of "The Future of Essentially Derived Variety (EDV) Status: Predominantly More Explanations or Essential Change"

_agronomy, doi:10.3390/agronomy11061261_

Round 1

Reviewer 1 Report

This is a timely and well-thought piece on essential derivation, and adds to the area at time when UPOV is revising its Explanatory Notes on EDVs.

Author Response

Thank you

no revisions appear to be necessary according to reviewer 1

Reviewer 2 Report

Thanks for the opportunity to review the article.

The work provides a useful contribution to the present discussions about Essentially Derived Varieties (EDV) at the International Union for the Protection of New Varieties of Plants (UPOV) and the deliberations of the Working Group on Essentially Derived Varieties (WG-EDV). 

For this reviewer, however, addressing the following matters would significantly improve the article: 

1.    The first sentence “The capacity of research and product development … thereby contributing to more productive crop production” has a very obscure meaning (maybe?). This sentence should be recast to that it is clear what the author actually means and the number of complex ideas liberated for the benefit of the readers. 

2.    Sui generis – This term is used in the first paragraph to suggest that plant variety intellectual property is “sui generis”. This is true in the sense that the World Trade Organisation’s Agreement on Trade Related Aspects of Intellectual Property (TRIP) requires a patent or “effective sui generis” plant variety system (Art 27.3(b)). But Plant Variety Protection (PVP) or Plant Breeders’ Rights (PBR) by itself is not sui generis to anything! Perhaps this might need clarification in the first paragraph. The point being that UPOV is an acceptable “effective sui generis” plant variety system for the purposes of TRIPS.

3.    The words “It is abundantly clear that” starting the second paragraph are not necessary. With great respect, some plant breeding has dramatically changed, while lots of plant breeding remains very much like it was a long time ago and well before 1961.

4.    Part 2 of the article “Aspects of IP in plant breeding and agriculture” is not really necessary for the substance of this article, and perhaps more importantly, sets out and overly brief overview with some significant errors in understanding. For examples: 

a.    Section 2.1 provides some unsubstantiated opinions about “IP protection” and “human behaviour”. With great respect, “human nature” is trivial and entirely irrelevant in tracing the origins of intellectual property to cultivated plants and confuses tangible and intangible property. These topics, often called “traditional knowledge” and “traditional cultural expressions” are important to intellectual property, but not to our modern conception of PVP and PBR under UPOV. 
b.    Section 2.3 provides “While modern usage of ‘intellectual property’ traces to the establishment of the World Intellectual Property Organization in 1967”. This is simply not true. The World Intellectual Property Organization (WIPO) traces back to the United International Bureaux for the Protection of Intellectual Property established in 1893 to administer the Berne Convention for the Protection of Literary and Artistic Works and the Paris Convention for the Protection of Industrial Property. With great respect, it was these conventions that were the “modern” origin of intellectual property. Mark Lemley unfortunately and incorrectly opines that in the United States intellectual property really only took shape with WIPO (Mark Lemley, ‘Property, Intellectual Property, and Free Riding Archived’ (2005) 83 Texas Law Review 1031, 1033 (footnote 4)) perhaps cementing the exceptionalism in the United States’ recent conversion to the ideals of intellectual property and overlooking its long history.
c.    Section 2.3 also quotes “‘the labors of the mind’ as being ‘as much a man's own ... as what he cultivates, or the flocks he rears’” and attributes this to Mark Lemley. With great respect, this quote was from William Davoll et al. vs. James S. Brown, 3 West.L.J. 151 (1845).

Perhaps start the section from “As of February 22nd 2021, UPOV has 77 members …”?

5.    The last paragraph of Section 2 starting “Most European plant breeders …” includes a perspective from the International Seed Federation. Does the author have some affiliation with this organisation and is this a veiled justification for that organisation’s points? If yes, then the author should disclose his interests. If not, then provide some justification for this partisan perspective and why we should care what the International Seed Federation thinks. The same concern applies to the repeated preference for the International Seed Federation over other voices. 

6.    Section 3.3 addresses a range of court cases concluding that they do not provide clear guidance. With great respect, these cases clearly show that depending on which party has the onus of proof the alternative outcome follows. It might have been useful to have engaged with some of the legal literature about those cases and some discussion about the technicality requirements as they have been addressed in those cases and commentaries as the author appears to favour (like the International Seed Federation) a technicality approach to assessing EDVs. 

7.    With great respect, referring to citations “[63-70]” does not reflect the huge literature about these matters. It might be better to acknowledge that this is has been a significant research focus for some time without actually delivering a comprehensive technical standard that can confidently be applied in assessing EDV standards. 

8.    Section 4.2 addresses an interpretation of Article 14(5)(b)(iii) and the author sets out a very special way of interpreting this provision. Surely, disease resistance is exactly the “except for the differences which result from the act of derivation”? 

9.    We can go all Aristotle defining objects by their essences, but as others have noted over the millennia, the essences of biological objects like plants are impossible to identify. The flaw is to assume this is merely a technical issue that can be resolved by a technical standard (such as the author’s favoured International Seed Federation suggests). 

10.    With great respect, the superficial rejection of the Australian approach to assessing EDV overlooks a critical problem of dealing with EDV as just a technical issue. The Sir Walter v B12 decision clearly shows that breeding a shorter internode was critical to the breeding a better grass and that the B12 was a superior variety. This was exactly the subjective kind of assessment required by standards like non-obviousness that go to the qualities of the breeding, rather than just breeding per se. 

11.    With great respect, the statement in Section 5.1 “It is technically incorrect to assert …” may well be true, but it is not very useful as EDV is all about similarities and enough difference to warrant a separate PVP/PBR. The author needs to actually engage with the technicality issue and that EDV might not actually be a technical question. There is literature out there addressing this point including the cited Sanderson (Law in Context), although that was cited in this article for another purpose. 

Author Response

I thank reviewer no 2 for their very good comments. I have responded positively to all the comments. These revisions have greatly improved the manuscript.

Thanks for the opportunity to review the article.

The work provides a useful contribution to the present discussions about Essentially Derived Varieties (EDV) at the International Union for the Protection of New Varieties of Plants (UPOV) and the deliberations of the Working Group on Essentially Derived Varieties (WG-EDV). 

GENERAL AUTHOR COMMENTS

The comments of reviewer 2 emphasize that I have spoken in this manuscript to the subject of EDV solely in the light of technical aspects. These comments prompt me to place the technical aspects within the larger context. Technical data can measure genetic similarities between varieties, which then relates to the degree of preponderance in derivation or inheritance. Furthermore, such data can help inform decisions on essential derivation by helping place them in context. For example, it is known from basic genetics and indeed has been validated by research into EDV that breeding strategies such as backcrossing (bc) are used with the intent of retaining certain %s of inheritance from a particular parent. On average, individuals developed from 1 bc preferentially retain, on average, 75% of the genetics that differ between the initial parents (the segregating genetics). The actual % genetic or marker similarity will be greater due to the genetics that is common among the parents.  With 2 backcrosses the mean rises to 87.5% of the segregating genetics. The % rises as numbers of bc generations increase. Research has also shown that there is a relatively large distribution around this mean. So, for example with 1 bc and marker-based selection one can find individual segregants that are in the 90%+ range of segregating genetics. The actual numerical data depend upon marker method, crop species, and relationship between the parents involved in making the initial breeding cross. These data can inform the decision making on an EDV threshold. These data can inform the decision takers with regard to the intent of the developer of the second variety. IE Was the intent to retain a predominant portion of the genetics of the initial variety?  Additional data including comparisons of morphological and agronomic data can inform the decision maker on EDV status by providing a metric on i) the characteristics of the initial variety that have been retained and ii) the number and extent of differences between the initial and second developed variety  that have resulted from the act of derivation (breeding). To be pertinent to help inform decision making these data will be required to meet certain standards, including on a crop species specific basis, using markers that are highly discriminative within the crop species and which are not “cherry picked” for a specific inquiry, are available to all parties, and are reliably useful. However, at the end of the day-the decision on essential derivation cannot itself be made on a purely technical basis not the least because the decision maker will have to interpret language, including of the UPOV Convention, UPOV Explanatory notes, and legal precedent. This language includes “predominant”, “essential”, and “a few”. Decision makers can include the parties involved, crop specific bodies, arbitration panels, and a court of law.

For this reviewer, however, addressing the following matters would significantly improve the article: 

  1. The first sentence “The capacity of research and product development … thereby contributing to more productive crop production” has a very obscure meaning (maybe?). This sentence should be recast to that it is clear what the author actually means and the number of complex ideas liberated for the benefit of the readers. 

Author response. I agree that the first sentence holds within it a number of ideas. Given that this manuscript is about the policy of essential derivation within the UPOV system of intellectual property it makes most sense to simply delete this sentence. The Introduction then dives immediately into the subject at hand.

  1. Sui generis – This term is used in the first paragraph to suggest that plant variety intellectual property is “sui generis”. This is true in the sense that the World Trade Organisation’s Agreement on Trade Related Aspects of Intellectual Property (TRIP) requires a patent or “effective sui generis” plant variety system (Art 27.3(b)). But Plant Variety Protection (PVP) or Plant Breeders’ Rights (PBR) by itself is not sui generis to anything! Perhaps this might need clarification in the first paragraph. The point being that UPOV is an acceptable “effective sui generis” plant variety system for the purposes of TRIPS.

Author response. The primary point I am making is that PVP is not a Utility Patent system. I have used text provided by the reviewer to explain this. I thank the reviewer for this clarification. I do understand the point that the reviewer makes and so I have added text to indicate that the WTO agreement on TRIPS regards the UPOV system as an acceptable “effective sui generis”. I also make the point that there are other sui generis regimes for the protection of plant varieties, e.g. those adopted in India, Thailand and Malaysia. Essential derivation is included in the Indian PVP Act.

  1. The words “It is abundantly clear that” starting the second paragraph are not necessary. With great respect, some plant breeding has dramatically changed, while lots of plant breeding remains very much like it was a long time ago and well before 1961.

Author response. The reviewer is entirely correct. I have revised the text including to state that plant breeding for many crops and in many regions remains very much like it was before 1961—whether. I deleted “It is abundantly clear that” and added  “although their impact has varied across crops and regions.”

To reflect the role of technical data in the larger process of determining essential derivation I have provided additional text in the Introduction:

Determination of essential derivation is highly dependent upon data that can inform a decision on the degree to which, i) one variety is inherited or derived from another, and ii) apart from changes introduced during further development, the extent to which essential characteristics of the initial variety have been retained in the derivative(s). A sound basis of technical information can both contribute to and help inform a decision on essential derivation. However, increased technological capabilities to transfer or change the expression of several genes simultaneously may further question the degree of genetic change and/or breeding effort required to warrant a commercial status that is independent of essential derivation. Consequently, it is appropriate to fundamentally question whether the introduction of essential derivation in 1991 remains a practical solution to support a balance of IP rights between initial and subsequent developers of new varieties.

  1.    Part 2 of the article “Aspects of IP in plant breeding and agriculture” is not really necessary for the substance of this article, and perhaps more importantly, sets out and overly brief overview with some significant errors in understanding. For examples: 
  2.    Section 2.1 provides some unsubstantiated opinions about “IP protection” and “human behaviour”. With great respect, “human nature” is trivial and entirely irrelevant in tracing the origins of intellectual property to cultivated plants and confuses tangible and intangible property. These topics, often called “traditional knowledge” and “traditional cultural expressions” are important to intellectual property, but not to our modern conception of PVP and PBR under UPOV. 
    b.    Section 2.3 provides “While modern usage of ‘intellectual property’ traces to the establishment of the World Intellectual Property Organization in 1967”. This is simply not true. The World Intellectual Property Organization (WIPO) traces back to the United International Bureaux for the Protection of Intellectual Property established in 1893 to administer the Berne Convention for the Protection of Literary and Artistic Works and the Paris Convention for the Protection of Industrial Property. With great respect, it was these conventions that were the “modern” origin of intellectual property. Mark Lemley unfortunately and incorrectly opines that in the United States intellectual property really only took shape with WIPO (Mark Lemley, ‘Property, Intellectual Property, and Free Riding Archived’ (2005) 83 Texas Law Review 1031, 1033 (footnote 4)) perhaps cementing the exceptionalism in the United States’ recent conversion to the ideals of intellectual property and overlooking its long history.
    c.    Section 2.3 also quotes “‘the labors of the mind’ as being ‘as much a man's own ... as what he cultivates, or the flocks he rears’” and attributes this to Mark Lemley. With great respect, this quote was from William Davoll et al. vs. James S. Brown, 3 West.L.J. 151 (1845).

Perhaps start the section from “As of February 22nd 2021, UPOV has 77 members …”?

Author response. Good discussion THANKS!  I do agree that the sections cited by the reviewer are not the focus of this review. I prefer to start section 2 a little earlier than proposed by the reviewer so as to provide references and a broad overview of UPOV. These edits mean that section 2 requires a new heading: 2. The UPOV approach to PVP. 

  1. The last paragraph of Section 2 starting “Most European plant breeders …” includes a perspective from the International Seed Federation. Does the author have some affiliation with this organisation and is this a veiled justification for that organisation’s points? If yes, then the author should disclose his interests. If not, then provide some justification for this partisan perspective and why we should care what the International Seed Federation thinks. The same concern applies to the repeated preference for the International Seed Federation over other voices. 

Author response. I was trying to indicate why Utility Patents can be regarded as not the most suitable form of IP for protection of varieties per se.

Given again that is subject is not the focus of this review I have deleted this paragraph. One could write another manuscript , indeed a book on this topic, in fact yet another publication might be redundant given those already published.

I was not in any way trying to make a veiled justification for ISF’s view. I was active in the ISF as an employee of Pioneer Hi-Bred 1980-2015 and indeed was chair of the IP committee of the ISF until I retired from that position in 2016. I have no affiliation with the ISF other than being able to attend their annual Congress without having to pay the registration fee. Neither do I have an affiliation with Corteva, the successor of Pioneer Hi-Bred, then DuPont Pioneer.

  1. Section 3.3 addresses a range of court cases concluding that they do not provide clear guidance. With great respect, these cases clearly show that depending on which party has the onus of proof the alternative outcome follows. It m ight have been useful to have engaged with some of the legal literature about those cases and some discussion about the technicality requirements as they have been addressed in those cases and commentaries as the author appears to favour (like the International Seed Federation) a technicality approach to assessing EDVs. 

Author Response. Excellent critique. As I have previously noted and made efforts to amend is that technical data can provide a sound basis to help a decision taker with regard to essential derivation. Molecular maker data are necessary to determine derivation. Marker data can help inform the decision maker on intent of the second developer. There are standards that should be applied in order to provide relevant and meaningful data to aid the decision maker(s). Nonetheless, a decision on EDV status can only be made consider technical evidence in respect of an interpretation of the language provided by UPOV, possible consideration any further information provided by crop specific groups, and/or judicial precedence. My contention is that the judicial decisions made to date have not been well served by the data from the molecular marker analyses that were available to the courts.

I have added text to make this point with some examples.     

  1. With great respect, referring to citations “[63-70]” does not reflect the huge literature about these matters. It might be better to acknowledge that this is has been a significant research focus for some time without actually delivering a comprehensive technical standard that can confidently be applied in assessing EDV standards. 

Author response. The reviewer makes an excellent point. I have added and revised the original text speaking to the requirements for molecular data to be useful evidential material rather than to set thresholds per se

  1. Section 4.2 addresses an interpretation of Article 14(5)(b)(iii) and the author sets out a very special way of interpreting this provision. Surely, disease resistance is exactly the “except for the differences which result from the act of derivation”?

Author response I absolutely agree! That is what I was trying to day BUT I see I had an error which likely caused the confusion: The error was

For example, to add the characteristic of disease resistance to a derived variety means that it no longer retains the essential characteristic of susceptibility to that disease that is a feature of the iv. Hence, use of the language in UPOV 1991: “except for the differences which result from the act of derivation”.

NOW corrected:

For example, to add the characteristic of disease resistance to an initial variety means that it no longer retains the essential characteristic of susceptibility to that disease that was a feature of the iv. Hence, use of the language in UPOV 1991: “except for the differences which result from the act of derivation”.

  1.    We can go all Aristotle defining objects by their essences, but as others have noted over the millennia, the essences of biological objects like plants are impossible to identify. The flaw is to assume this is merely a technical issue that can be resolved by a technical standard (such as the author’s favoured International Seed Federation suggests). 

Author response: OK I agree that this is not ONLY a technical issue. My point is though that unless the technical questions on degree and thresholds of similarity cannot be resolved then determinations of EDV status whether by arbitration, tribunal or court will have a weak and inadequate evidential basis for determination of essential derivation. In several places in the revised text I have emphasized the point that marker data can help in determining predominant derivation and they can help inform decision makers on the intent of the second breeder. I make the point that marker data per se cannot be the sole determinant of EDV status, that is unless a group has decided that is how EDV status can be resolved. Even in those circumstances I would presume that such a decision could still be taken to a court of law for further judicial review and determination.  

  1.    With great respect, the superficial rejection of the Australian approach to assessing EDV overlooks a critical problem of dealing with EDV as just a technical issue. The Sir Walter v B12 decision clearly shows that breeding a shorter internode was critical to the breeding a better grass and that the B12 was a superior variety. This was exactly the subjective kind of assessment required by standards like non-obviousness that go to the qualities of the breeding, rather than just breeding per se. 

Author response: This is not a technical problem. It’s an interpretation that flys in the face of the original intent of essential derivation. I entirely agree that B12 is a superior variety in terms of withstanding tread wear. The initial question regarding EDV is – IS B12 so similar to the initial variety Sir Walter that it might be an EDV. Australia then decided B12 is NOT an EDV because it has an economically valuable characteristic even though it is otherwise apparently the same as Sir Walter. This adoption of economic value to escape EDV status takes Australia back to UPOV 1978—it’s as simple as that. The EDV concept was NOT introduced just to deal with plagiarism or cosmetic breeding. If it took a lot of breeding effort to develop B12 during the protection period of Sir Walter then a proposal to replace the EDV approach with another approach makes sense.

  1.    With great respect, the statement in Section 5.1 “It is technically incorrect to assert …” may well be true, but it is not very useful as EDV is all about similarities and enough difference to warrant a separate PVP/PBR. The author needs to actually engage with the technicality issue and that EDV might not actually be a technical question. There is literature out there addressing this point including the cited Sanderson (Law in Context), although that was cited in this article for another purpose. 

Author response

OK, Yes, the title of this subsection is way too broad!

In addition to earlier edits to the text to indicate the relative role of technical data.

I hope that my additional text regarding lessons from the generation and presentation of molecular marker data from previous cases addresses the fair criticism: “The author needs to actually engage with the technicality issue and that EDV might not actually be a technical question.”

 I have edited the text in a similar fashion here also:

5.1. White Paper on “Essentially Derived Varieties.” [73]

Clearly, decision makers, including of course the judiciary will  be those who make the determination of essential derivation. Technical data require their generation, presen-tation , and interpretation. A comprehensive set of technical information provides eviden-tial material that can be drawn upon to help in the determination of EDV status. However, proposals that rely primarily on phenotype with use of a compulsory license [73] are overly limited. This approach is flawed by i) requiring all essential characteristics to be retained, ii) removing the initial prerequisite of determining predominant derivation, and iii) being imbalanced by placing an over-reliance upon phenotypic compared to genotypic data.

Reviewer 3 Report

Please find comments and suggestions for Authors in the attached word file.

Author Response

I thank reviewer no 3 for their very good critique. I have responded positively to all the comments. The revisions have improved the manuscript significantly.

EDV MS Reviewer 3 Comments AND AUTHOR RESPONSES

Review Report for The Future of Essentially Derived Variety (EDV) Status: Predominantly More Explanations or Essential Change Journal: Agronomy (ISSN 2073-4395) Manuscript: ID agronomy-1243102 Date of submission: 8 June 2021

This manuscript attempts to undertake a comprehensive account of how UPOV 1991 and its member countries may regulate the development and use of Essentially Derived Varieties (EDVs). In so doing, the manuscript covers several key aspects of how global debates about essential derivation are progressing, also identifying very relevant questions and concerns associated with the development and use of the concept of EDVs. A very innovative part of the manuscript is the discussion of a new proposal that UPOV, and perhaps other relevant parties may consider for the regulation of EDVs, both internationally and nationally.

 For making this manuscript publishable, I provide the following comments. First, the manuscript needs extensive editing. Most sentences are too long, and in some cases, there are grammatical errors as well. The manuscript needs to be edited for clarity and coherence as well.

Author Response. I have broken up several of the longer sentences.

 Second, some sections of the manuscript are irrelevant. For example, Section 2. This section attempts to provide a brief historical account of how seed systems, plant breeding, and IP as a law evolved, but no direct linkage between this section and other sections has been established throughout the manuscript. Furthermore, the discussion of informal seed exchange practices in this section has been done without showing why this issue matters for EDVs. I recommend the deletion of this section.

Author response. I agree, this section has been deleted.

The last three concluding paragraphs in the last section of the manuscript also may be deleted or edited. Especially the last paragraph on page 12 does not provide a good concluding statement to this manuscript.

Author response. I can agree, deleted

Third, the manuscript lacks discussion of how countries have taken diverse approaches to regulate EDVs and what insights can be drawn from these approaches to inform UPOV negotiations on EDVs. There is a bit of discussion of the Australian PVP system (page 7), but that is too short and may not be sufficient. Countries such as India have already registered and provided protection to a range of EDVs. These developments may provide some insights into how in practice countries are classifying and protecting EDVs. This manuscript may include one or two case studies of why and with what criteria and requirements, EDVs have been registered.

Author response

I have revised the text relating to the Australian interpretation of essential derivation. I have tried to present a more balanced view. Nonetheless, the end result of this interpretive route is to i) provide additional strong resistance to plagiarism while, ii) undermining the whole concept of balanced protection for the initial breeder and a subsequent developer who makes a relatively small genetic change-which, yes may indeed have high economic value. One can indeed argue, legally and economically why this makes sense-as I have heard Doug Waterhouse do at UPOV in Geneva. However,  if that is the case then, in my mind,  either the current guidelines for essential derivation need to be radically changed, or a revised UPOV Convention is required.

I have added text, including that the Indian PPVRFA including to the unique ability to apply for protection as an EDV  that thereby helps protect all developers including farmer-breeders given that extant variety is also a class for protection.

I have added a section:

3.4 Technical practices involved in the implementation of essential derivation according to the PPVFRA of India.

I have added text speaking to the issues surrounding AFLP data in the Gypsophila cases in the Netherlands and in Israel.

Fourth, there is a statement on page 10: “when germplasm is under the terms of the International Treaty or the CBD, then a portion of license fees could accrue back to those providers.” This statement holds true in the case of the CBD, but not for the International Treaty. The International Treaty on Plant Genetic Resources for Food and Agriculture has a distinct mechanism of benefit sharing. According to this Treaty, all monetary payments must be made to an international benefit sharing fund and not to the germplasm providers (e.g., see Adhikari, Kamalesh (2018). Reconceptualizing access: moving beyond the limits of international biodiversity laws. Biodiversity, genetic resources and intellectual property: developments in access and benefit sharing. (pp. 9-32) edited by Charles Lawson and Kamalesh Adhikari. New York, NY United States: Routledge. doi: 10.4324/9781315098517-2)

Author response Yes indeed, my mistake-an oversight. I want to include both the Global Crop Diversity Trust and the benefit sharing fund since both are acknowledged as supporting the IT. Revised text is:

when germplasm is used under the terms of the International Treaty or the CBD, then a portion of license fees could accrue back to support the goals of the Treaty via the Global Crop Diversity Trust and the benefit sharing fund of the Treaty or to other providers, if accessed under the CBD.

Round 2

Reviewer 3 Report

Thanks for addressing my previous comments.